# Thai *Curcuma* Species: Antioxidant and Bioactive Compounds

**DOI:** 10.3390/foods9091219

**Published:** 2020-09-02

**Authors:** Supawadee Burapan, Mihyang Kim, Yingyong Paisooksantivatana, Bekir Engin Eser, Jaehong Han

**Affiliations:** 1Metalloenzyme Research Group and Department of Plant Science and Technology, Chung-Ang University, Anseong 17546, Korea; i_chi60@yahoo.com; 2Phytobean, AC. Ltd., Pori 2-gil 16-5, Gamcheon-myeon, Yechon 36810, Korea; mihcoterie@gmail.com; 3Department of Horticulture, Faculty of Agriculture, Kasetsart University, Bangkok 10900, Thailand; yp2624@yahoo.com; 4Department of Engineering, Aarhus University, Gustav Wieds Vej 10, 8000 Aarhus, Denmark; bekireser@eng.au.dk

**Keywords:** antioxidant, bioactive, *Curcuma*, curcuminoids, HPLC, sesquiterpenoids, total phenolic content

## Abstract

For the functional food applications, antioxidant properties and the bioactive compounds of the 23 *Curcuma* species commercially cultivated in Thailand were studied. Total phenolic content and DPPH radical scavenging activity were determined. The concentrations of eight bioactive compounds, including curcumin (**1**), demethoxycurcumin (**2**), bisdemethoxycurcumin (**3**), 1,7-diphenyl-(4*E*,6*E*)-4,6-heptadien-3-ol (**4**), germacrone (**5**), furanodienone (**6**), zederone (**7**), and *ar*-turmerone (**8**), were determined from the *Curcuma* by HPLC. While the total phenolic content of *C. longa* was highest (22.3 ± 2.4 mg GAE/g, mg of gallic acid equivalents), *C.* Wan Na-Natong exhibited the highest DPPH (2,2-diphenyl-1-picryl-hydrazyl-hydrate) radical scavenging activity. Twenty-three *Curcuma* species showed characteristic distributions of the bioactive compounds, which can be utilized for the identification and authentication of the cultivated *Curcuma* species. *C. longa* contained the highest content of curcumin (**1**) (304.9 ± 0.1 mg/g) and *C. angustifolia* contained the highest content of germacrone (**5**) (373.9 ± 1.1 mg/g). It was noteworthy that 1,7-diphenyl-(4*E*,6*E*)-4,6-heptadien-3-ol (**4**) was found only from *C. comosa* at a very high concentration (300.7 ± 1.4 mg/g). It was concluded that Thai *Curcuma* species have a great potential for the application of functional foods and ingredients.

## 1. Introduction

Turmeric, *Curcuma longa*, is the only *Curcuma* species extensively cultivated and traded in the world [1]. Taxonomically, it belongs to the *Curcuma* genus of the Zingiberaceae family, and is mostly distributed in Asia [2,3]. Various beneficial health-promoting effects of *C. longa*, such as Alzheimer′s disease prevention, anti-inflammatory effects, and HIV-1 protease inhibition, were reported [4,5]. Curcuminoids and sesquiterpenoids were identified as the major bioactive constituents of *C. longa* [6]. More than a hundred species of *Curcuma* are reported worldwide, and about 30 species among them are cultivated and consumed in Thailand as food additives, cosmetics, and traditional medicines [7]. In particular, many of them are used for the treatment of various diseases due to the existence of bioactive compounds. However, bioactive compounds in other *Curcuma* species, other than *C. longa*, have never been studied extensively.

To expand the application of *Curcuma* species as functional foods, we have collected and cultivated 23 Thai *Curcuma* species widely consumed in Thailand. Total phenolic content and antioxidant activity were measured in the ethanol extracts of dried *Curcuma* rhizomes. For the bioactive compound study, four curcuminoids and four sesquiterpenoids were isolated and used as reference compounds after complete characterizations, because these are popularly studied representative phytochemicals in *Curcuma* species. In detail, the curcuminoids curcumin (**1**), demethoxycurcumin (**2**), and bisdemethoxycurcumin (**3**) were isolated from *C. longa*, and the other curcuminoid 1,7-diphenyl-(4*E*,6*E*)-4,6-heptadien-3-ol (**4**) was isolated from *C. comosa*. The three germacrane-type sesquiterpenoids germacrone (**5**), furanodienone (**6**), and zederone (**7**) were isolated from *C. latifolia*, and *ar*-turmerone (**8**) was isolated from *C. zedoaria* (Figure 1). The compositions of these representative bioactive compounds in the *Curcuma* species were also analyzed by HPLC.

## 2. Materials and Methods

### 2.1. Curcuma Species

The rhizomes of 23 *Curcuma* species were collected in Kanchanaburi in 2013 and cultivated in Tapong, Meaung Rayong, Thailand in 2014 throughout the year. The plants that formed flowers were processed to prepare depository specimens at Bangkok Herbarium, Plant Variety Protection Division, Department of Agriculture, Bangkok, Thailand.

### 2.2. Plant Materials and Extraction

Fresh rhizomes of the 23 *Curcuma* species were rinsed several times with tap water to make them dust and debris free. The rhizomes were then cut into small pieces and dried at room temperature in the shade for 2–3 days. The dried samples were ground to a fine powder and kept in a deep freeze (−70 °C) for further experiments. For extraction, each sample (1 g) of *Curcuma* sp. was macerated in EtOH (20 mL) for 24 h. The ethanol extract was filtered and dried under reduced pressure. The residue was weighed for the extraction yield and dissolved in MeOH for the determination of total phenolic content and DPPH radical scavenging assay.

### 2.3. Determination of Total Phenolic Contents

Total phenolic contents of all plant extracts were determined using Folin–Ciocalteu reagent as described by Singleton and Rossi [8]. The plant extracts were dissolved in MeOH (1 mL, 0.5 mg/mL). Samples were added to microtiter plates and mixed with 100 µL of a 10-fold diluted Folin–Ciocalteu reagent and 80 µL of 7.5% sodium carbonate. The absorbance at 765 nm was measured using microplate reader spectrophotometers (Spectramax190, Molecular Device, CA, USA) after 30 min. Total phenolic content was expressed as mg GAE/g. The calibration curve of standard gallic acid solution was obtained as *y* = 0.005 *x* (A_765_) + 0.081, R^2^ = 0.998, where *y* = mg GAE/g, at the region between 5 and 200 µg/mL.

### 2.4. DPPH Radical Scavenging Assay 

The antioxidant activity of the *Curcuma* extracts was evaluated by DPPH radical scavenging. The extract in MeOH (2 mg/mL) and ascorbic acid in MeOH (200 µg/mL) were prepared. The test solution (50 µL) was mixed with 950 µL of DPPH (0.1 mM in MeOH). After 30 min, the absorbance at 516 nm was measured using spectrophotometer (UV/visible detector) in triplicate. DPPH radical scavenging activity was calculated using the following formula: % scavenging = 100 × ((absorbance of control–absorbance of sample test)/absorbance of control).

### 2.5. Isolation of Compounds

The dried rhizomes (500 g) of *C. longa* were ground and extracted with EtOH (1 L) by maceration (24 h). The dried crude extracts (5 g) were suspended in water and extracted by EtOAc (500 mL × 3). The combined organic layer (500 mg) was isolated by vacuum liquid chromatography using hexanes and an acetone gradient. Out of the 36 fractions, fractions 25–34 (150 mg) were combined and rechromatographed by column chromatography (4% MeOH in chloroform) to afford 40 fractions. Curcumin (**1**) was isolated from fractions 9–14 (30 mg), demethoxycurcumin (**2**) was isolated from fractions 19–23 (15 mg), and bisdemethoxycurcumin (**3**) was isolated from fractions 25–35 (55 mg).

Curcumin (**1**). Yellow powder; ESI *m/z* 369 [M + H]^+^; m.p. 181–183 °C; Anal. calcd for C_21_H_20_O_6_: C 68.47, H 5.47, found: C 67.12, H 5.47. ^1^H NMR (600 MHz, DMSO-*d*_6_): δ 7.54 (2H, *d*, *J* = 15.8 Hz, H-4, 4′), 7.32 (2H, *d*, *J* = 2.0 Hz, H-6, 6′), 7.15 (2H, *dd*, *J* = 8.3 Hz, 2.0 Hz, H-10, 10′), 6.82 (2H, *d*, *J* = 8.1 Hz, H-9, 9′), 6.75 (2H, *d*, *J* = 15.8 Hz, H-3, 3′), 6.06 (1H, *s*, H-1), 3.84 (6H, *s*, 7-OCH_3_, 7′-OCH_3_). ^13^C NMR (151 MHz, DMSO-*d*_6_): δ 183.26 (C-2, 2′), 149.43 (C-8, 8′), 148.06 (C-7, 7′), 140.76 (C-4, 4′), 126.37 (C-5, 5′), 123.21 (C-10, 10′), 121.13 (C-3, 3′), 115.76 (C-9, 9′), 111.35 (C-6, 6′), 100.91 (C-1), 55.76 (7-OCH_3_, 7′-OCH_3_).

Demethoxycurcumin (**2**). Yellow orange powder; ESI *m/z* 339 [M + H]^+^; m.p. 170–171 °C; Anal. calcd for C_20_H_18_O_5_: C 70.99, H 5.36, found: C 68.53, H 5.17. ^1^H NMR (600 MHz, DMSO-*d*_6_): δ δ 10.0 (1H, *br s*, 8-OH), 7.56 (2H, *d*, *J* = 8.7 Hz, H-6, 10), 7.57 (2H, *dd*, *J* = 15.8 Hz, 5.3 Hz, H-4, 4′), 7.31 (1H, *d*, *J* = 2.0 Hz, H-6′), 7.14 (1H, *dd*, *J* = 8.2 Hz, 2.0 Hz, H-10′), 6.82 (3H, *dm*, H-7, 9, 9′), 6.75 (H, *d*, *J* = 16 Hz, H-3′), 6.69 (H, *d*, *J* = 16 Hz, H-3), 6.04 (1H, *s*, H-1), 3.51 (1H, *s*, H-1′), 3.83 (3H, s, 7′-OCH_3_). ^13^C NMR (125 MHz, DMSO-*d*_6_): δ 183.69 (C-2), 183.56 (C-2′), 160.25 (C-8), 149.81 (C-8′), 148.43 (C-7′), 141.13 (C-4′), 140.79 (C-4), 130.76 (C-6, C-10), 126.76 (C-5′), 126.24 (C-5), 123.63 (C-10′), 121.46 (C-3′), 121.24 (C-3), 116.34 (C-7, C-9), 116.12 (C-9′), 111.68 (C-6′), 101.32 (C-1), 56.13 (OMe).

Bisdemethoxycurcumin (**3**). Yellow orange crystals; ESI *m/z* 309 [M + H]^+^; m.p. 220–221 °C; Anal. calcd for C_19_H_16_O_5_: C 74.01, H 5.23, found: C 69.72, H 5.51. ^1^H NMR (600 MHz, DMSO-*d*_6_): δ 10.1 (2H, *br s*, 8,8′-OH), 7.57 (4H, *dd*, *J* = 6.6 Hz, 2.1 Hz, H-6, 6′, 10, 10′), 7.54 (2H, *d*, *J* = 16 Hz, H-4, 4′), 6.82 (4H, *dm*, *J* = 8.6 Hz, H-7, 7′, 9, 9′), 6.70 (2H, *d*, *J* = 16 Hz, H-3, 3′), 6.04 (1H, *s*, H-1), 3.51 (1H, *s*, H-1′). ^13^C NMR (151 MHz, DMSO-*d*_6_): δ 183.32 (C-2, 2′), 159.88 (C-8, 8′), 140.48 (C-4, 4′), 130.44 (C-6, 6′, 10, 10′), 125.93 (C-5, 5′), 120.90 (C-3, 3′), 116.04 (C-7, 7′, 9, 9′), 101.05 (C-1).

The dried rhizome (100 g) of *C. comosa* was ground and macerated with MeOH (200 mL) for 24 h. The dried crude extracts (2 g) were partitioned by EtOAc and water (300 mL × 3). The EtOAc extract (300 mg) was further isolated by vacuum liquid chromatography using hexanes and acetone. Fractions 12–15 gave **4** (48 mg). 1,7-Diphenyl-(4*E*,6*E*)-4,6-heptadien-3-ol (**4**) Pale yellow solid; ESI *m/z* 247 [M + H]^+^; Anal. calcd for C_19_H_18_O: C 86.99, H 6.92, found: C 86.37, H 7.59. ^1^H NMR (300 MHz, CDCl_3_): δ 7.27 (2H, *br*, H-3′, H-5′), 6.75 (2H), 5.83 (H-4), 4.21 (1H, s, OH), 2.71 (H-7).

The sesquiterpenoids were isolated from the ethanol extract of *C. latifolia* by the same procedure as for curcuminoids. Germacrone (**5**) was isolated from fractions 9–10 (18 mg), furanodienone (**6**) was isolated from fraction 15 (10 mg), and zederone (**7**) was isolated from fractions 20–22 (38 mg).

Germacrone (**5**). White crystal; EI *m/z* 218 [M]^+^; m.p. 55–56 °C; Anal. calcd for C_15_H_22_O: C 82.52, H 10.16, found: C 80.81, H 9.83. ^1^H NMR (600 MHz, CDCl_3_): δ 4.99 (1H, *br d*, *J* = 12.2 Hz, H-1), 4.71 (1H, *br d*, *J* = 11.2 Hz, H-5), 3.41 (1H, *d*, *J* = 10.6 Hz, H-9a), 2.99–2.83 (3H, *m*, H-6a, H-6b, H-9b), 2.37 (1H, m, H-2a), 2.19–2.05 (3H, *m*, H-2b, H-3a, H-3b), 1.78 (3H, *s*, 13-CH_3_), 1.73 (3H, *s*, 12-CH_3_), 1.63 (3H, *s*, 14-CH_3_), 1.44 (3H, *s*, 15-CH_3_). ^13^C NMR (151 MHz, CDCl_3_): δ 207.92 (C-8), 137.25 (C-11), 135.00 (C-4), 132.68 (C-1), 129.48 (C-7), 126.68 (C-10), 125.38 (C-5), 55.91 (C-9), 38.09 (C-3), 29.23 (C-6), 24.09 (C-2), 22.34 (C-13), 19.90 (C-12), 16.72 (C-14), 15.59 (C-15).

Furanodienone (**6**). Yellow pale oil; EI *m/z* 230 [M]^+^; m.p. 84–86 °C; Anal. calcd for C_15_H_18_O_2_: C 78.23, H 7.88, found: C 64.84, H 6.91. ^1^H NMR (600 MHz, CDCl_3_): δ 7.08 (1H, *m*, H-12), 5.81 (1H, *m*, H-5), 5.18 (1H, *br dd*, *J* = 11.7 Hz, 4.3 Hz, H-1), 3.70 (2H, *br d*, *J* = 22 Hz, H-9), 2.47 (1H, *dt*, *J* = 11.4 Hz, 3.6 Hz, H-3a), 2.32 (1H, *m*, H-2a), 2.18 (1H, *m*, H-2b), 2.13 (3H, *d*, *J* = 1.2 Hz, 13-CH_3_), 2.00 (3H, *d*, *J* = 1.2 Hz, 15-CH_3_), 1.89 (1H, *m*, H-3b), 1.31 (3H, bd, J = 0.7 Hz, 14-CH_3_). ^13^C NMR (151 MHz, CDCl_3_): δ 189.81 (C-6), 156.52 (C-8), 145.79 (C-4), 138.07 (C-12), 135.39 (C-10), 132.44 (C-5), 130.51 (C-1), 123.71 (C-11), 122.18 (C-7), 41.70 (C-3), 40.66 (C-9), 26.44 (C-2), 18.98 (C-15), 15.78 (C-14), 9.55 (C-13).

Zederone (**7**). White crystal; EI *m/z* 246 [M]^+^; m.p. 149–151 °C; (69.13 %C, 7.11 %H) Anal. calcd for C_15_H_18_O_3_: C 73.15, H 7.37, found: C 69.13, H 7.11. ^1^H NMR (600 MHz, CDCl_3_): δ 7.08 (1H, *m*, H-12), 5.48 (1H, *m*, H-1), 3.81 (1H, *s*, H-5), 3.75 (1H, *d*, *J* = 16.3 Hz, H-9a), 3.69 (1H, *d*, *J* = 16.3 Hz, H-9b), 2.52 (1H, *m*, H-2a), 2.30 (1H, *dt*, *J* = 13.1 Hz, 3.5 Hz, H-3a), 2.26–2.20 (1H, *m*, H-2b), 2.11 (3H, *d*, J = 1.3 Hz, 13-CH_3_), 1.34 (3H, *d*, *J* = 0.8 Hz, 15-CH_3_), 1.32–1.25 (1H, *m*, H-3b). ^13^C NMR (151 MHz, CDCl_3_): δ 192.19 (C-6), 157.08 (C-8), 138.05 (C-12), 131.19 (C-1), 131.04 (C-10), 123.24 (C-11), 122.21 (C-7), 66.54 (C-5), 63.96 (C-4), 41.88 (C-9), 37.98 (C-3), 24.64 (C-2), 15.72 (C-14), 15.14 (C-15), 10.26 (C-13).

*ar*-Turmerone (**8**) was isolated from *C. zedoaira*, and 65 mg of **8** was obtained from 100 g of the dried rhizomes. *ar*-Turmerone (**8**). Colorless oil; EI *m/z* 216 [M]^+^; Anal. calcd for C_15_H_20_O: C 83.28, H 9.32, found: C 79.31, H 8.82. ^1^H NMR (600 MHz, CDCl_3_): δ 7.10 (4H, aromatic, H), 6.02 (1H, *J* = 1.3 Hz), 3.28 (1H, *dd*, *J* = 8.1 Hz), 2.70 (1H, *dd*, *J* = 15.6 Hz), 2.60 (1H, *dd*, *J* = 15.6 Hz), 2.31 (3H, *s*, -CH_3_), 2.10 (3H, *d*, *J* = 1.3 Hz, -CH_3_), 1.85 (3H, *d*, *J* = 1.3 Hz, -CH_3_), 1.25 (3H, *d*, *J* = 1.3 Hz, 12-CH_3_). ^13^C NMR (151 MHz, CDCl_3_): δ 199.85, 155.06, 143.68, 135.53, 129.10, 126.65, 124.08, 52.68, 35.28, 27.63, 21.98, 20.97, 20.70.

### 2.6. Validation of HPLC Analysis

Validation of HPLC analysis was performed using the purified reference compounds. Calibration curves were constructed from the HPLC peak areas of the reference standards, **1**–**8**, versus their concentrations. Linearity was calculated by measuring the eight points of the calibration curve in the range of 0.003125-0.4 mM (**1**–**3**) and 0.00625-0.8 mM (**4**–**8**) with triplicate measurements. The limits of detection (LOD) for **1**–**8** were found at the micromolar level. Demethoxycurcumin (**2**) and furanodienone (**6**) showed the lowest LOD (2.0 μM) and limits of quantification (LOQ) at 420 nm and 245 nm, respectively (Table 1).

### 2.7. Compositional Analysis of Curcuma Species 

Each sample (1 g) of the *Curcuma* species was macerated in EtOH (20 mL) for 24 h. The extract was filtered and dried under reduced pressure. The dried residue was dissolved in DMF (*N*,*N*-dimethylformamide, 1 mg/1 mL) for the analysis with a Finnigan Surveyor Plus HPLC system. The mobile phase comprised 0.1% acetic acid in water (solvent A) and 0.1% acetic acid in MeCN (solvent B). The injection volume for HPLC analysis was 10 μL and the flow was 1.0 mL/min. For the quantitative analysis, UV absorption at 420 nm was adopted for **1**–**3**, and UV absorption at 245 nm was adopted for **4**–**8**.

### 2.8. Statistical Analyses

All measurements were performed in triplicate. Data were processed by Microsoft Excel and reported as means ± standard deviation.

## 3. Results and Discussion

### 3.1. Taxonomy of Curcuma Species

Among the 23 Curcuma species, 11 Curcuma species, such as *C. manga*, *C. aeruginosa*, *C. comosa*, *C. aurantiaca*, *C. aromatic*, *C. latifolia*, *C. zedoaria*, *C. longa*, *C. parviflora*, *C. angustifolia*, and *C. petiolate*, were identified by the taxonomist at the Bangkok Herbarium. The other 12 species are new species, and designated by common names in this report (Table 2).

### 3.2. Total Phenolic Content and Antioxidant Activity

Total phenolic contents of the *Curcuma* species varied from 0.4 ± 0.1 to 22.3 ± 2.4 mg GAE/g. The rhizomes of *C. longa* contained the highest phenolic contents (22.3 ± 2.4 mg GAE/g), followed by *C. parfivlora* (15.3 ± 1.2 mg GAE/g) and *C. latifolia* (12.9 ± 0.3 mg GAE/g). The DPPH radical scavenging activity of *Curcuma* extracts is also presented in Table 2. *Curcuma* Wan Na-Natong showed the highest DPPH radical scavenging activity (91.8 ± 0.6%), followed by *C. comosa* (90.0 ± 0.3%) and *Curcuma* Wan Chai-Dam (89.8 ± 0.6%). These three species showed very strong antioxidant activity, even stronger than *C. longa*, and require further study.

From the results, it was clear that total phenolic contents of *Curcuma* extracts are not necessarily correlated with the antioxidant activity (Figure 2). Among the three most antioxidant *Curcuma* rhizomes, *C.* Wan Na-Natong and *C.* Wan Chai-Dam are the new species that have never been extensively studied, regardless of their local popularity. The correlation chart (Figure 2) shows that the total phenolic content and DPPH radical scavenging antioxidant activity among the *Curcuma* species are not necessarily linearly correlated. This is believed to be due to different phytochemical compositions of *Curcuma* species, as discussed below.

### 3.3. Isolation of Bioactive Curcuminoids and Sesquiterpenoids

Curcuminoids **1**–**3** were isolated from *C. longa*, and **4** was isolated from *C. comosa*. Sesquiterpenoids **5**–**7** were isolated from *C. latifolia* for the first time in this study. Compound **8** was isolated from *C. zedoaria*. The structures of the isolated reference compounds (Figure 1) were confirmed by the comparison of melting point, elemental analysis, ^1^H, and ^13^C NMR data (see the Appendix A) [9,10,11,12,13,14,15,16].

### 3.4. Chemical Composition of 23 Curcuma Species

Curcuminoids **1**–**3** are considered as the most characteristic bioactive secondary metabolites found in turmeric. Therefore, we analyzed the collected *Curcuma* species to find out whether the rhizomes contain **1**–**3**. The EtOH extracts (1 mg/1 mL) were analyzed with HPLC at 420 nm [17,18], and only ten species were found to contain curcuminoids (Table 3). Furthermore, only three *Curcuma* species, *C. zedoaria*, *C. longa*, and *Curcuma* Wan Khamintong, contained all three curcuminoids (Figure 3). Trace amounts of curcuminoids were identified from *C. manga, Curcuma* Wan Rang-Jud, *C. comosa, C. latifolia, C. parviflora, Curcuma* Wan Pataba, *Curcuma* Wan Na-Natong, and *C. petiolate* under LOQ, only when the higher concentrations of the analytes were analyzed by HPLC. It is noticeable that *C. longa* was found to contain the highest curcuminoid content (653 mg/g). Even though curcumin (**1**) is the representative bioactive compound of turmeric, only nine *Curcuma* species contained curcumin (**1**). Demethoxycurcumin (**2**) was the major curcuminoid of four other *Curcuma* species, *C. aeruginosa*, *C. aurantiaca*, *C. aromatica*, and *C. zedoaria*.

From Indonesian *Curcuma* species, *C. mangga*, *C. heyneana*, *C. aeruginosa*, and *C. soloensis* were reported to contain curcuminoids [19]. Our analysis also confirmed that *C. mangga* and *C. aeruginosa* contained curcuminoids. *C. aurantiaca* contained high contents of **1**–**3**, and this is the first report of the identification of **1**–**3** in the rhizomes of *C. aurantiaca*. Diarylheptanoid 1,7-diphenyl-(4*E*,6*E*)-4,6-heptadien-3-ol (**4**) belongs to the curcuminoids, but it was found only from *C. comosa* in a high content of 300.7 ± 1.4 mg/g. It is noteworthy that *C. comosa* does not contain **1**–**3**.

There are more than 100 different sesquiterpenoids reported from *Curcuma* rhizomes [20], and these secondary bioactive metabolites are also responsible for some of the biological activities observed from the rhizomes of *Curcuma* species. Therefore, we isolated three germacrane-type sesquiterpenoids, germacrone (**5**), furanodienone (**6**), and zederone (**7**), which is the first report from *C. latifolia*. In addition, *ar*-turmerone (**8**) was isolated from *C. zedoaria* for the first time. Germacrone (**5**) is known to exhibit anti-inflammatory [21], antiviral [22], and anti-cancer activities [23], and was found in the various *Curcuma* species, such as *C. aeruginosa*, *C. amanda*, *C. aromatica*, *C. xanthrrhiza*, and *C. zedoaria*.

Among the 23 *Curcuma* species analyzed, 21 species were found to contain at least one of the four sesquiterpenoids, and only two *Curcuma* species, *C. aurantiaca* and *C. zedoaria*, contained all four sesquiterpenoids (Table 4). Generally, germacrone (**5**) was the most abundant sesquiterpenoid in the *Curcuma* species, and *C. angustifolia* contained only **5**, out of the eight reference compounds. In particular, *C. angustifolia* showed a very high concentration of **5** (37% of extract), and *Curcuma* Wan Khamin-Khao-Padtalod contained only *ar*-turmerone (**8**).

## 4. Conclusions

The phytochemical property of 23 *Curcuma* species was studied by means of total phenolic contents and antioxidant activity assays, as well as compositional analysis. Besides, eight bioactive curcuminoids and sesquiterpenoids were isolated from the rhizomes of *C. longa*, *C. comosa*, *C. latifolia*, and *C. zedroaria*. In particular, the isolation of germacrone (**5**), furanodienone (**6**), and zederone (**7**) is the first report from *C. latifolia*. From the compositional analysis, *C. longa* was distinct in its highest curcumin (**1**) concentration. *C. angustifolia* was found to contain the highest content of germacrone (**5**). Out of 23 *Curcuma* species, five *Curcuma* species did not contain any curcuminoids **1**–**4**. None of the analyzed *Curcuma* species contained all eight bioactive compounds. Therefore, the data of phytochemical profiling can be used for the identification and authentication of cultivated *Curcuma* species. In addition, there is a tremendous potential for the utilization of less-studied *Curcuma* species as functional foods and ingredients.

## Figures and Tables

**Figure 1 foods-09-01219-f001:**
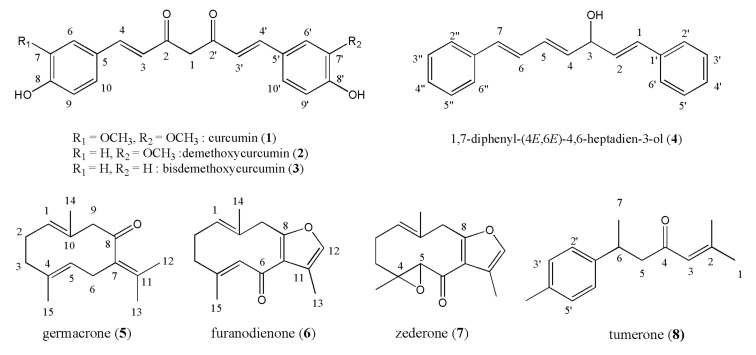
Structure of curcuminoids and sesquiterpenoids isolated from *Curcuma* species.

**Figure 2 foods-09-01219-f002:**
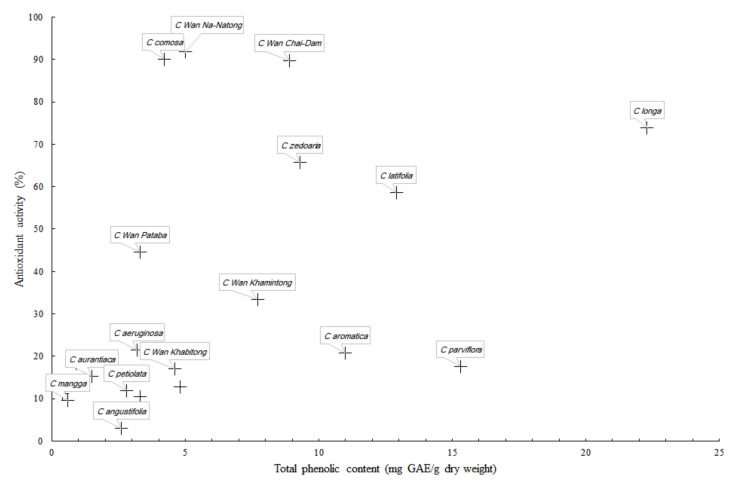
Correlation chart of total phenolic content and DPPH radical scavenging antioxidant activity. The data are from Table 2, and only notable data are labeled in the chart.

**Figure 3 foods-09-01219-f003:**
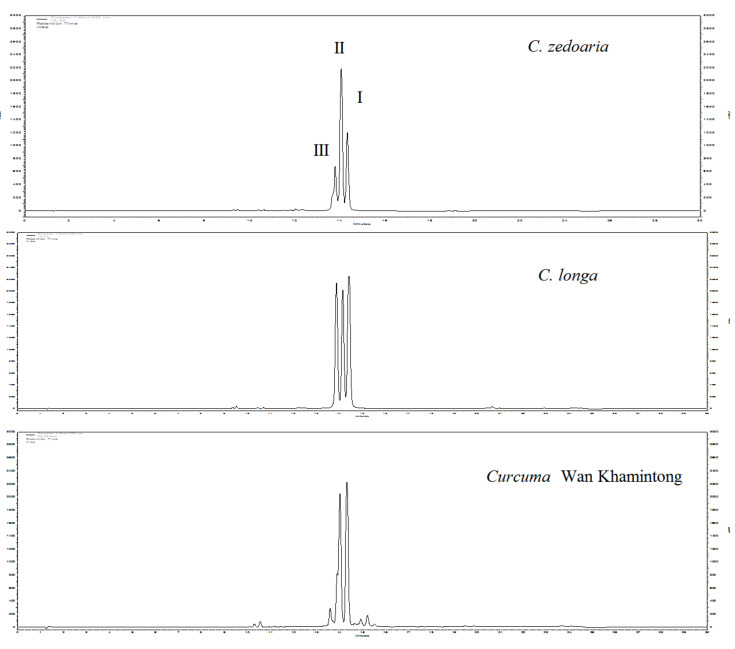
HPLC chromatogram (420 nm) for *C. zedoaria*, *C. longa*, and *Curcuma* Wan Khamintong; I: curcumin (**1**), II: demethoxycurcumin (**2**), III: bisdemethoxycurcumin (**3**).

**Table 1 foods-09-01219-t001:** Validation of HPLC assay of curcuminoids and sesquiterpenoids.

Compounds	Accuracy ^1^	Linearity	Correlation Coefficient (R²)	LOD (mM)	LOQ (mM)
Curcumin (1)	104.8 ± 7.9	*y* = 56,277,482*x* + 476,424	0.9951	0.0081	0.0245
Demethoxycurcumin (2)	101.0 ± 7.9	*y* = 50,455,350*x* + 56,885	0.9997	0.0020	0.0061
Bisdemethoxycurcumin (3)	96.5 ± 16.2	*y* = 54,929,237*x* + 369,859	0.9964	0.0069	0.0209
1,7-diphenyl-(4*E*,6*E*)-4,6-heptadien-3-ol (4)	106.9 ± 11.6	*y* = 28,237,427*x* + 612,834	0.9963	0.0125	0.0377
Germacrone (5)	97.5 ± 16.6	*y* = 3,051,741*x* + 22,600	0.9977	0.0110	0.0333
Furanodienone (6)	101.9 ± 6.8	*y* = 8,043,402x − 10,111	0.9999	0.0020	0.0060
Zederone (7)	99.9 ± 5.1	*y* = 5,130,378*x* + 25,981	0.9999	0.0026	0.0080
*ar*-Turmerone (8)	98.5 ± 15.0	*y* = 35,723,577*x* + 423,877	0.9962	0.0125	0.0379

^1^ All values are presented as the mean ± SD of triplicate determinations.

**Table 2 foods-09-01219-t002:** Total phenolic content and antioxidant activity of *Curcuma* species.

Curcuma Species	Extraction Yield (%)	Total Phenolic Content (mg GAE/g Dry Weight)	Antioxidant Activity (%) ^1^
*Curcuma* Wan Ma-Leung	6.5 ± 0.4	0.9 ± 0.0	18.2 ± 0.3
*Curcuma* *mangga*	6.9 ± 0.3	0.6 ± 0.3	9.6 ± 0.5
*Curcuma* Wan Ma-Hor	8.9 ± 0.2	0.5 ± 0.1	13.2 ± 0.5
*Curcuma* Wan Khamin-Dam	13.5 ± 0.2	3.9 ± 0.4	20.4 ± 0.4
*Curcuma* Wan Rang- Jud	6.3 ± 0.3	0.4 ± 0.1	11.2 ± 0.2
*Curcuma* *aeruginosa*	8.6 ± 0.2	3.2 ± 0.3	21.5 ± 0.3
*Curcuma comosa*	16.8 ± 0.3	4.2 ± 0.1	90.0 ± 0.3
*Curcuma* Wan Kanta-Mala	12.0 ± 1.0	4.8 ± 0.1	12.7 ± 0.3
*Curcuma* *aurantiaca*	11.9 ± 0.1	1.5 ± 0.1	15.2 ± 0.4
*Curcuma* *aromatica*	18.5 ± 0.3	11.0 ± 0.2	20.8 ± 0.1
*Curcuma latifolia*	20.3 ± 0.2	12.9 ± 0.3	58.6 ± 0.0
*Curcuma* *zedoaria*	8.5 ± 0.6	9.3 ± 0.7	65.7 ± 0.2
*Curcuma* *longa*	21.6 ± 0.5	22.3 ± 2.4	73.9 ± 0.1
*Curcuma* *parviflora*	11.4 ± 0.5	15.3 ± 1.2	17.5 ± 0.5
*Curcuma* *angustifolia*	3.2 ± 0.3	2.6 ± 0.3	3.0 ± 0.2
*Curcuma* Wan Khabitong	11.7 ± 0.1	4.6 ± 0.1	17.0 ± 0.1
*Curcuma* Wan Pataba	5.8 ± 0.2	3.3 ± 0.1	44.5 ± 0.8
*Curcuma* Wan Kortong	8.4 ± 0.5	3.3 ± 0.1	10.4 ± 0.4
*Curcuma* Wan Na-Natong	3.7 ± 0.2	5.0 ± 0.1	91.8 ± 0.6
*Curcuma* *petiolata*	5.5 ± 0.1	2.8 ± 0.4	11.8 ± 0.4
*Curcuma* Wan Khamin-Khao-Padtalod	6.2 ± 0.3	1.9 ± 0.8	7.7 ± 0.2
*Curcuma* Wan Chai-Dam	3.9 ± 0.2	8.9 ± 0.2	89.8 ± 0.6
*Curcuma* Wan Khamintong	8.6 ± 0.2	7.7 ± 1.3	33.3 ± 0.7

^1^ Antioxidant activity (%): final concentration of sample = 100 µg/mL and ascorbic acid final concentration = 10 µg/mL (95.1 ± 0.2).

**Table 3 foods-09-01219-t003:** Analysis of curcuminoids from the *Curcuma* species.

Curcuma Species	1	2	3
*Curcuma* Wan Ma-Leung	5.2 ± 0.1	4.1 ± 0.0	ND ^1^
*Curcuma* Wan Ma-Hor	ND	0.2 ± 0.0	ND
*Curcuma aeruginosa*	35.5 ± 0.7	107.2 ± 1.0	ND
*Curcuma* *aurantiaca*	3.2 ± 0.0	6.5 ± 0.0	ND
*Curcuma aromatica*	24.3 ± 0.1	112.5 ± 1.0	ND
*Curcuma zedoaria*	72.3 ± 0.6	201.5 ± 1.5	28.2 ± 0.8
*Curcuma longa*	304.9 ± 0.1	189.2 ± 0.4	158.8 ± 0.7
*Curcuma* Wan Khabitong	10.2 ± 0.1	6.6 ± 0.0	ND
*Curcuma* Wan Kortong	0.1 ± 0.0	1.8 ± 0.0	ND
*Curcuma* Wan Khamintong	47.2 ± 0.1	37.1 ± 0.1	8.0 ± 0.0

^1^ ND = in trace below detection limit.

**Table 4 foods-09-01219-t004:** Analysis of sesquiterpenoids from the ethanol extraction residues of *Curcuma* species.

*Curcuma* Species	Sesquiterpenoids (mg/g) ^1^			
	5	6	7	8
*Curcuma* Wan Ma-Leung	42.6 ± 0.9	ND ^2^	ND	ND
*Curcuma* Wan Ma-Hor	47.5 ± 0.7	4.0 ± 0.0	6.6 ± 0.2	ND
*Curcuma* Wan Khamin-Dam	126.5 ± 0.3	18.5 ± 0.1	ND	ND
*Curcuma aeruginosa*	ND	3.1 ± 0.0	ND	ND
*Curcuma comosa*	31.1 ± 0.2	81.0 ± 0.9	37.6 ± 0.2	ND
*Curcuma* Wan Kanta-Mala	106.9 ± 0.4	15.7 ± 0.1	ND	ND
*Curcuma aurantiaca*	36.5 ± 0.4	15.4 ± 0.5	8.6 ± 0.1	4.9 ± 0.0
*Curcuma aromatica*	73.7 ± 2.8	4.7 ± 0.0	33.2 ± 0.6	ND
*Curcuma latifolia*	56.2 ± 0.3	73.3 ± 1.1	61.3 ± 0.1	ND
*Curcuma zedoaria*	7.5 ± 0.1	15.5 ± 0.4	3.3 ± 0.0	53.9 ± 0.2
*Curcuma longa*	ND	ND	ND	78.4 ± 0.2
*Curcuma parviflora*	ND	3.6 ± 0.1	ND	ND
*Curcuma angustifolia*	285.3 ± 0.6	ND	ND	ND
*Curcuma* Wan Khabitong	6.8 ± 0.1	ND	ND	4.4 ± 0.2
*Curcuma* Wan Patab	31.2 ± 0.3	2.5 ± 0.0	ND	0.2 ± 0.0
*Curcuma* Wan Kortong	ND	ND	ND	3.5 ± 0.2
*Curcuma* Wan Na-Natong	144.3 ± 2.8	47.1 ± 0.4	13.2 ± 0.1	ND
*Curcuma petiolata*	ND	ND	ND	0.4 ± 0.0
*Curcuma* Wan Khamin-Khao-Padtalod	ND	ND	ND	17.6 ± 0.1
*Curcuma* Wan Chai-Dam	ND	65.1 ± 0.2	13.5 ± 0.3	ND
*Curcuma* Wan Khamintong	ND	ND	ND	4.6 ± 0.0

^1^ Based on the weight of ethanol extraction residue from dried *Curcuma* species (wt%). ^2^ ND = not detected.

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
