# Peer review of "Thai Curcuma Species: Antioxidant and Bioactive Compounds"

_foods, 2020, doi:10.3390/foods9091219_

Round 1

Reviewer 1 Report

The work presented for review is very interesting and valuable due to the pioneering research of new species of Curcuma. It has been shown that two previously unexplored species have a higher antioxidant potential than the well-known and commonly used C. longa. Below are the comments.

line 61:  please specify drying conditions (e.g., temperature)

line 63: what was the alcohol concentration?

line 64: why ethanol was used for maceration and why the extract was dissolved in methanol to determine the antioxidant activity. Ethanol can also be used. What was the reason for this?

line 65: what was the freezing temperature?

lines 40-50: This section is more relevant to the Methodology or Results than to the Introduction.

- in how many repetitions total phenolic contents have been determined?

line 83 and line 112: were the samples shaken during the maceration? What was the volume of alcohol?

- In part „Isolation of compounds” methods are given for only two species C. longa i C. comosa- why ? For these two species different amounts of sample and alcohol were used, why were they different?

line 160: what was the alcohol concentration?

line 161: please explain the abbreviation „DMF”

lines 184-187: based on the results obtained, the correlation coefficient between these parameters can be calculated

Table 2: It would be valuable to supplement the information whether the values of antioxidant potential and polyphenols content between different Curcuma species were statistically significant

Author Response

Thank you for the constructive comments. We made changes and improved the manuscript according to the comments. Details responses are found below.

Review 1

The work presented for review is very interesting and valuable due to the pioneering research of new species of Curcuma. It has been shown that two previously unexplored species have a higher antioxidant potential than the well-known and commonly used C. longa. Below are the comments.

line 61:  please specify drying conditions (e.g., temperature)

  • “at room temperature” was added to clarify.

line 63: what was the alcohol concentration?

  • Without % means usually absolute (100%) ethanol. So didn’t make any changes.

line 64: why ethanol was used for maceration and why the extract was dissolved in methanol to determine the antioxidant activity. Ethanol can also be used. What was the reason for this?

  • It is a good point. There is no reason to use different solvents for the different steps of experiments. However, we adopted the current experimental conditions because these are commonly used in the literatures.

line 65: what was the freezing temperature?

  • The temperature, -70°C, was added in line 66.

lines 40-50: This section is more relevant to the Methodology or Results than to the Introduction.

  • The introduction part was improved by updating the styles and references, though we believe brief introduction of methodology in Introduction part is appropriate.

- in how many repetitions total phenolic contents have been determined?

  • Three times, as mentioned at the end of Materials and Methods.

line 83 and line 112: were the samples shaken during the maceration? What was the volume of alcohol?

  • The volumes were added in the manuscript at L88 and L117.

- In part „Isolation of compounds” methods are given for only two species C. longa i C. comosa- why ? For these two species different amounts of sample and alcohol were used, why were they different?

  • Based on the preliminary analysis, two species were selected and the extraction solvents were chosen by the target molecules.

line 160: what was the alcohol concentration?

  • Without % means usually absolute (100%) ethanol. So didn’t make any changes.

line 161: please explain the abbreviation „DMF”

  • Full name was added.

lines 184-187: based on the results obtained, the correlation coefficient between these parameters can be calculated

  • No apparent linear correlation was found between total phenolic content and antioxidant activity, as described in the manuscript. The coefficient was not calculated, but new Figure showing the relationship was added.

Table 2: It would be valuable to supplement the information whether the values of antioxidant potential and polyphenols content between different Curcuma species were statistically significant

  • Thank for the helpful comments. The correlation chart was newly added as Figure 2.

Reviewer 2 Report

This communication reports the phytochemical analysis of Curcuma species obtained in Thailand. Although the novelty of this research is limited, the results may be valuable as an accumulation of phytochemical information on Curcuma species.

Please consider the following suggestions:

Line 18
The abbreviations “GAE” and “DPPH” must be defined. These abbreviations are popular, but I think that not all readers know.

Line 43
Please explain why you selected the four curcuminoids and four sesquiterpenoids as representative bioactive compounds in this study.

Figure 1
The letter “E” (4E) in “1,7-diphenyl-(4E,6E)-4,6-heptadien-3-ol (4)” must be italicized.
Compounds 4, 7, and 8 have asymmetric carbons. If possible, please show the stereochemistry.

Lines 55–56
The year and month of the cultivation must be described.

Lines 91–148
All these compounds have been reported previously. These data are unnecessary. Instead, you must cite appropriate papers, and just describe that the data from purified compounds coincided with those of the papers. If you would like to show these data, please move them to the Supplementary.

Author Response

Thank you for the constructive comments. We made changes and improved the manuscript according to the comments. Details responses are found below.

 Review 2

This communication reports the phytochemical analysis of Curcuma species obtained in Thailand. Although the novelty of this research is limited, the results may be valuable as an accumulation of phytochemical information on Curcuma species.

Please consider the following suggestions:

Line 18
The abbreviations “GAE” and “DPPH” must be defined. These abbreviations are popular, but I think that not all readers know.

  • Revised as suggested.

Line 43
Please explain why you selected the four curcuminoids and four sesquiterpenoids as representative bioactive compounds in this study.

  • The compounds were chosen, because these are popularly studied phytochemicals in Curcuma species. L46-47.

Figure 1
The letter “E” (4E) in “1,7-diphenyl-(4E,6E)-4,6-heptadien-3-ol (4)” must be italicized.
Compounds 4, 7, and 8 have asymmetric carbons. If possible, please show the stereochemistry.

  • Typo was corrected. For the chiral center, we would like to keep the current structure because the absolute configuration was not confirmed by ourselves.

Lines 55–56
The year and month of the cultivation must be described.

  • The information was added in L58-59.

Lines 91–148
All these compounds have been reported previously. These data are unnecessary. Instead, you must cite appropriate papers, and just describe that the data from purified compounds coincided with those of the papers. If you would like to show these data, please move them to the Supplementary.

  • We agree your opinion in part. However, all the data available from the literature cited are outdated and obtained from the old models of instruments. Therefore, we would like to keep the current format for the readers.